# Advancements in Research on Constructing Physiological and Pathological Liver Models and Their Applications Utilizing Bioprinting Technology

**DOI:** 10.3390/molecules28093683

**Published:** 2023-04-24

**Authors:** Zibei Ming, Xinyu Tang, Jing Liu, Banfeng Ruan

**Affiliations:** School of Biology, Food and Environment, Hefei University, Hefei 230601, China; mzb1031@163.com (Z.M.); txy111110@163.com (X.T.)

**Keywords:** 3D bioprinting, bioink, drug testing, tissue engineering, liver

## Abstract

In recent decades, significant progress has been made in liver tissue engineering through the use of 3D bioprinting technology. This technology offers the ability to create personalized biological structures with precise geometric design capabilities. The complex and multifaceted nature of liver diseases underscores the need for advanced technologies to accurately mimic the physiological and mechanical characteristics, as well as organ-level functions, of liver tissue in vitro. Bioprinting stands out as a superior option over traditional two-dimensional cell culture models and animal models due to its stronger biomimetic advantages. Through the use of bioprinting, it is possible to create liver tissue with a level of structural and functional complexity that more closely resembles the real organ, allowing for more accurate disease modeling and drug testing. As a result, it is a promising tool for restoring and replacing damaged tissue and organs in the field of liver tissue engineering and drug research. This article aims to present a comprehensive overview of the progress made in liver tissue engineering using bioprinting technology to provide valuable insights for researchers. The paper provides a detailed account of the history of liver tissue engineering, highlights the current 3D bioprinting methods and bioinks that are widely used, and accentuates the importance of existing in vitro liver tissue models based on 3D bioprinting and their biomedical applications. Additionally, the article explores the challenges faced by 3D bioprinting and predicts future trends in the field. The progress of 3D bioprinting technology is poised to bring new approaches to printing liver tissue in vitro, while offering powerful tools for drug development, testing, liver disease modeling, transplantation, and regeneration, which hold great academic and practical significance.

## 1. Introduction

The World Health Organization (WHO) reports that liver diseases, including the hepatitis B and C viruses (HBV and HCV), cirrhosis, liver cancer, fatty liver, and drug-induced liver injury, affect 1.5 billion individuals globally [1]. Unfortunately, liver disease causes over 2 million deaths annually, which is steadily increasing [2]. A recent publication in the International Journal of Epidemiology predicts that without intervention, the incidence and death rate of liver cancer in China will continue to rise until 2030, with the most significant increase seen in steatohepatitis-related liver cancer [3]. Currently, the clinical treatment options for liver diseases include drug therapy, interventional therapy, and surgical treatment; drug therapy is the most common and least intrusive treatment for liver disease [4]. However, the challenge of developing new drugs is significant, as many liver disorders lack targeted and effective medications. Traditionally, new drugs are explored through two-dimensional cell culture models and animal testing. However, the complex and multistage nature of liver disease [5] makes it difficult to fully understand the various phases of the disease and to test drugs using these limited models. Although two-dimensional cell culture models [6] are affordable and easy to operate, they present limitations, such as difficulty maintaining long-term cultures, susceptibility to cellular dedifferentiation, and an inability to accurately represent the dynamic metabolic environment of drugs in the human body. The use of animal models is widespread in liver disease research; however, they come with challenges, including ethical concerns, lengthy procedures, and high expenses. Additionally, significant differences exist between the animal and human metabolic capacities and physicochemical microenvironments, making the results obtained from animal studies less relevant [7]. Given these limitations, liver tissue engineering and drug research require a new, alternative approach that enables a longer-term culture of liver tissues to study the causes of liver disease and evaluate potential drugs.

3D bioprinting technology enables precise placement of biomaterials and living cells and control over the shape, structure, and size of the printed materials, allowing for the reconstruction of highly complex tissues and organs [8]. The resulting 3D structures can be customized to meet individual needs. Since patient cells are used, there is a low risk of rejection, making these structures suitable for direct implantation into the human body [9]. With the growing popularity of personalized patient treatment and precision medicine, the potential benefits of 3D bioprinting are becoming increasingly apparent [10]. Bioprinting offers significant biomimetic advantages in simulating the physiological characteristics and functions of liver tissue within the body [11]. This technology can provide new ideas and methods for exploring the biological characteristics and disease mechanisms of liver tissue and for the treatment, prevention, and drug screening of liver diseases.

As a result, the development of bioprinting could pave the way for creating new models that offer a more accurate representation of human physiology and help to bridge the gap between animal and human testing, which can be applied in tissue engineering, organ transplantation, and regenerative medicine. In Figure 1, an overview of the procedural steps for constructing liver tissue and its application using 3D bioprinting is presented. This process begins with the extraction of diverse liver cells from the liver, which are then cultured. Subsequently, the cultured cells are mixed with biomaterials to create the bioink, which is then used for 3D bioprinting liver tissue models using various printing techniques. The resulting liver tissue models are then cultured in vitro to generate physiological or pathological models for drug discovery and disease mechanism research.

## 2. Three-Dimensional Bioprinting Technology

The introduction of 3D bioprinting as a highly innovative manufacturing technique addresses the limitations of conventional manufacturing methods, making it one of the most thrilling and pioneering fields of the 21st century, especially in the field of biomedical research [12]. Regenerating and reconstructing damaged tissues and organs through organ and tissue regeneration techniques can serve to replace, restore, maintain, or enhance their functions [13]. The success of organ tissue engineering hinges on the ability to recreate three-dimensional structures that closely mimic human organs [14]. The utilization of 3D bioprinting technology enables the creation of organ tissue scaffolds with precise spatial geometry, microstructure, and mechanical properties. These characteristics are crucial in the fabrication of functional tissue constructs, making 3D bioprinting a promising tool in tissue engineering and regenerative medicine [15,16]. Depending on the specific application, various bioprinting technologies can be employed to achieve the desired results. As a complex and large organ, the liver requires a combination of bioprinting methods for its successful reproduction. Extrusion-based 3D bioprinting, for example, is highly flexible and efficient for printing liver lobules. However, for the intricate dual circulatory system of the liver, high-resolution, high-precision light-assisted 3D bioprinting might be preferred. Standard bioprinting methods include extrusion-based 3D bioprinting, inkjet-based 3D bioprinting, and light-assisted 3D bioprinting [17,18,19]. These methods can be used individually or in combination to build organ tissue through additive fabrication. Table 1 summarizes the main printing technologies used in 3D bioprinting.

### 2.1. Extrusion-Based 3D Bioprinting

The most widely used bioprinting technology in scientific research and tissue engineering is currently extrusion-based 3D bioprinting. Through a computer, users can create the desired shape and set related instructions [20]. The printer utilizes pneumatic or mechanical driving based on pre-designed commands, and the nozzle moves through specific 2D patterns or 3D structures to distribute bioink according to the desired shape. The printed cylindrical filaments are transformed into mechanically durable structures through ultraviolet light, enzymes, chemical substances, or thermal crosslinking [21]. Wu et al. [22] proposed a mixed bioink consisting of alginate and cellulose nanocrystals (CNCs) to enhance initial shape fidelity, which has excellent shear-thinning performance and can be easily squeezed through the nozzle. This bioink has been shown to print 3D liver tissue models containing fibroblast and liver cancer cells. This printing method can handle high-viscosity bioink and control the real-time flow of high-viscosity materials by adjusting the pressure and printing speed, allowing for linear printing of high-viscosity materials. However, one drawback of this printing method is the potential for shear stress inside the nozzle due to the pressure used to extrude the bioink, which may cause cell damage and reduced cell viability [23].

### 2.2. Inkjet-Based 3D Bioprinting

The use of inkjet-based 3D bioprinting technology is prevalent in the printing industry. What sets it apart from the usual 2D inkjet printers we are accustomed to is the utilization of biological ink. While working, the process of transforming “biological ink” into droplets and expelling them from a nozzle onto a receiving substrate occurs, resulting in the gradual formation of a 3D structure layer by layer. This is achieved by manipulating a combination of heat, piezoelectricity, electromagnetic force, and electric force [24]. A single droplet of biological ink can contain many cells, ranging from 10^4^ to 10^6^ [25]. The technology of inkjet 3D bioprinting has gained widespread usage among researchers, who employ it to print livers and blood vessels with great success. Boland et al. [26] demonstrated the potential of inkjet-based 3D bioprinting by mixing vascular endothelial cells with gelatin/alginate hydrogel using inkjet printing technology. Scanning electron microscopy analysis revealed that the endothelial cells grew within the hydrogel scaffold, maintaining excellent cell viability and growth status. Compared to 3D bioprinting methods that rely on extrusion, inkjet-based 3D bioprinting employs exceedingly delicate nozzles to print fluid materials through a jetting process. This technique offers simplicity in operation, affordability, speed, and remarkable material adaptability [27].

Additionally, in 3D bioprinting that utilizes inkjet technology, several nozzles can function concurrently, enabling high-capacity bioprinting. However, it has some limitations, such as a tendency for the ejected liquid to splatter, clogging of the nozzle, and lower precision. The accuracy of inkjet-based 3D bioprinting can only reach approximately one-tenth of extrusion-based 3D bioprinting [28].

### 2.3. Light-Assisted 3D Bioprinting

Light-assisted 3D bioprinting typically employs two printing methods: DLP-based and laser-based, both well-suited for printing high-precision, high-resolution structures. Using laser scanning or projection to crosslink photopolymerizable hydrogels containing cells at specific locations, solidification occurs, forming a robust structure [29,30]. Light-assisted 3D bioprinting can greatly enhance printing accuracy and resolution compared to extrusion-based and Inkjet-based 3D bioprinting. It can also swiftly produce a range of intricate structural patterns. Ma et al. [31] developed an in vitro model using photopolymerization to embed hiPSC-HPCs, human umbilical vein endothelial cells, and adipose-derived stem cells into a hexagonal liver lobule structure. Following several weeks of in vitro culture, the study demonstrated enhanced biological functions of hiPSC-HPCs, improved tissue and structure, increased expression levels of liver-specific genes, higher metabolite secretion, and enhanced cytochrome P450 induction. This model is suitable for early-stage drug testing and disease modeling (Figure 2a). With its high processing accuracy, non-contact processing, low environmental restrictions, strong adaptability, and high operability, light-assisted 3D bioprinting offers many advantages [32]. Nonetheless, this printing method can lead to phototoxicity and hinder cell growth, necessitating the use of specific photosensitive materials for the bioink used in the printing process [33]. The range of biological materials that can be used is limited.

**Table 1 molecules-28-03683-t001:** Concise summary of common bioprinting technologies in liver tissue engineering.

	Extrusion-Based	Inkjet-Based	Light-Assisted
Working Principle	The printer utilizes pneumatic or mechanical driving based on predesigned commands to move the nozzle and distribute bioink according to the desired shape.	The bioink is expelled from the nozzle in droplet form by adjusting the pressure, like a conventional 2D desktop inkjet printer.	Using laser scanning or projection to crosslink photopolymerizable hydrogels containing cells at specific locations, solidification occurs, forming a robust structure.
Advantages	It can handle high-viscosity bioink, allowing for linear printing of high-viscosity materials.	This technique offers simplicity in operation, affordability, speed, and remarkable material adaptability.	It offers high processing precision, non-contact processing, minimal environmental restrictions, adaptability, and operability.
Drawbacks	The generation of potential shear stress within the nozzle has the potential to cause cell damage and reduce cell viability.	It is easy to generate spattering from the expelled fluid, which leads to waste, and its accuracy is roughly one-tenth that of extrusion printing.	This method may lead to phototoxicity, which can hinder cell growth.
Print speed	Slow	Fast	Fast
Printer cost	Medium	Low	High
Cell density	Medium, 10^6^–10^7^	Low, 10^4^–10^6^	Medium, 10^7^
Ref.	[17,21,22,23]	[22,25,26,28]	[29,30,33]

## 3. Liver Tissue Printing Materials

### 3.1. Bioink

Bioink refers to a solution of biomaterials that is loaded with cells, which is essential for the ex vivo development of organ tissues in current 3D bioprinting technology. Bioink not only needs to help fix cells in the right place to form microstructures similar to those in the human body, supporting cell adhesion and proliferation [20,38], but also needs to support the physical demands of the printing process to withstand mechanical and thermal stresses during printing. Bioinks can be formulated using both natural and synthetic polymers. Natural polymers derived from natural sources are highly desirable, as they best mimic the characteristics of natural tissues. They possess similar properties as the extracellular matrix structure and exhibit good biocompatibility, biodegradability, and low cytotoxicity, and they support stable cell growth and proliferation; therefore, they are extensively utilized in bioprinting applications. Gelatin, alginate, collagen, decellularized liver extracellular matrix, hyaluronic acid, chitosan, and several other materials [39] are natural polymers that are widely applied in fields such as organ transplantation and regeneration. Commonly used synthetic polymers in 3D bioprinting include poly(ethylene glycol) diacrylate (PEGDA) [40], polyethylene glycol (PEG) [41], polycaprolactone (PCL) [42], polyvinylpyrrolidone (PVP) [43], and polyhydroxybutyrate (PHB) [44]. During printing, synthetic polymers can be tailored to provide essential biochemical and rheological properties and mechanical support and performance as softer materials and complex structures by adjusting their physical and chemical properties accordingly. Overall, natural polymers have inferior mechanical properties and cannot sustain structural integrity in the body for prolonged periods, leading to low cell support and a high risk of collapse [45]. Synthetic polymers may have higher cytotoxicity, lack natural cell attachment sites, and fail to mimic the properties of natural tissues accurately, which can negatively impact cell survival and proliferation [46].

Therefore, a single type of bioink is insufficient to meet the demands of complex structures. Natural and synthetic polymers are often combined during printing to obtain structures with high stability, resolution, fidelity, and biocompatibility [41]. In the upcoming section, we will delve into the distinctive features of five frequently encountered natural polymers and a commonly used synthetic polymer.

#### 3.1.1. Gelatin

Gelatin, a water-soluble protein derived from the natural polymer collagen, is widely known for its biocompatibility and biodegradability [47]. In 3D bioprinting, it acts as an extracellular matrix to support the physical and chemical functions of cells, and its degradation does not impact cell growth. Methacrylated gelatin (GelMA) can be produced through a reaction between gelatin and methacrylic anhydride; it has the proper viscosity to withstand extrusion printing and excellent biocompatibility and is commonly used for extrusion bioprinting [48]. Leach et al. [49] found that GelMA, with stiffness comparable to a healthy liver, supports cellular migration and proliferation. The combination of GelMA with triglyceride methacrylate hyaluronic acid (GMHA) creates a biocompatible microenvironment for endothelial cells and promotes their proliferation and vascularization [50]. However, the potential cytotoxicity of photocurable printing and the photo initiators used in GelMA-based bioinks need further investigation [51].

#### 3.1.2. Alginate

Alginates have gained widespread use as solid-phase scaffolds for immobilizing cells in various biomedical fields [52], thanks to their ease of availability, affordability, excellent moldability, and sufficient biocompatibility. Sodium alginate, acting as a sacrificial material, can be entirely eliminated after the printing process, thus giving rise to the formation of an internal structure or channel [52]. Sodium alginate has a drawback of low bioactivity, which may impede cell survival and proliferation; to address this, it can be combined with other biomaterials [53]. When combined with gelatin, alginate can form hydrogels that closely mimic the structure and composition of the natural extracellular matrix (ECM) [54]. This combination has become increasingly popular for bioprinting complex organs and multilayered vessels. However, one of the significant limitations of alginate is its weak mechanical properties, making it challenging to maintain its structure during long-term in vitro culture [55].

#### 3.1.3. Collagen

Collagen plays a crucial role as a key component of the extracellular matrix, making it the most popular material for cell scaffolds and tissue engineering due to its exceptional biocompatibility and low immunogenicity; these characteristics are vital for promoting cell growth and proliferation. However, collagen’s mechanical properties may not be ideal for printing applications, and as a result, it is often blended with various synthetic biomaterials to enhance the structural integrity and hardness of the final printed product [56]. Lee et al. [34] tackled this issue by mixing collagen with PCL, resulting in a framework that maintained the collagen’s shape and structure, while promoting the co-culture of three different types of cells. The PCL framework prevented collapse, enabling long-term experiments (Figure 2b).

#### 3.1.4. dECM

The extraction of decellularized matrix from animal liver tissues has gained popularity as a material for 3D bioprinting in recent years due to its good biocompatibility and ability to provide a biomimetic biochemical environment for cells [57]. This material retains the intact structure of the liver, allowing for the high viability and proliferation of hepatocytes over time, as well as the expression of relevant factor proteins and specific biological behaviors. Decellularized matrix (dECM) has been widely investigated as a bioink material, with numerous studies demonstrating its potential [58]. Kim et al. [59] developed a novel dECM bioink with improved cell viability and stem cell differentiation compared to conventional gelatin bioinks. Yu et al. [35] used dECM bioink in a DLP printing process to produce liver lobule structures for human pluripotent stem cell (hiPSC)-derived cardiomyocytes and hepatocytes, achieving high viability and maturation (Figure 2c).

#### 3.1.5. Fibrin

Fibrin is formed from soluble fibrinogen present in the blood [60]. Thrombin-induced proteolysis of the protein results in the formation of fibrin monomers, which assemble in both lateral and longitudinal directions to form a fibrillar network [61]. The fiber morphology of fibrin can be adjusted by changing the concentration ratio of thrombin and fibrinogen [62]. Fibrin exhibits inherent angiogenic properties and provides a stable environment for vascular formation. The fibrillar structure of the material acts as a scaffold for invading cells, which attach to its fibers via cellular receptors and develop capillaries. Wang et al. [63] proposed using fibrin hydrogel to construct a perfusable and hierarchical vascular network for liver tissue implantation; the cells achieved an exceptionally high survival rate following a six-day cultivation period. Due to this gel’s poor shape fidelity and mechanical properties, it is often mixed with other biomaterials, such as gelatin, sodium alginate, collagen, and hyaluronic acid, for various printing purposes [64].

#### 3.1.6. PEG

Polyethylene glycol (PEG) is a commonly used bioink in synthetic polymers due to its water solubility, low viscosity, and crosslinking properties [65]. However, PEG lacks good biocompatibility and cannot provide cell attachment sites for cell adhesion and proliferation, as natural tissues do. Therefore, it is often combined with natural polymers, and the biochemical properties can be adjusted, depending on the natural polymer used, to improve the mechanical properties and printing applicability [66]. PEG has been used to develop in vitro liver structures. Skardal et al. [67] introduced a novel hydrogel system composed of hyaluronic acid, gelatin, and PEG, which was used to bioprint liver spheroids to create an in vitro liver structure with high cell viability and tissue stiffness. PEG is used in combination with various materials in this new hydrogel system to provide mechanical support for printing structures.

### 3.2. Cell Sources

The complex signaling and metabolic environment in the liver are formed by the interaction of autocrine and paracrine signals among its cells [68], which drive the various activities of the liver. The following provides an overview of the different cell types and their functions in the liver.

#### 3.2.1. Hepatocytes

Hepatocytes are specialized epithelial cells that play a crucial role in endogenous and exogenous substance metabolism. These complex metabolic factories contain numerous organelles, including mitochondria, Golgi complexes, lysosomes, microsomes, and endoplasmic reticulum, which enable the execution of intricate metabolic processes [69]. Primary human hepatocytes (PHHs) freshly isolated from the human body are the optimal cell source for early drug development and in vitro liver tissue model construction [69,70]. PHHs more accurately represent the characteristics of the liver in vivo compared to other liver cell lines. Nguyen et al. [71] developed a novel bioprinted liver model using inkjet printing technology. In this model, primary hepatocytes, hepatic stellate cells (HSCs), and hepatic sinusoidal endothelial cells (HUVECs) were co-cultured to evaluate the toxicity of clinical drugs. Histological analysis showed diverse cell-to-cell connections between hepatocytes, desmin-positive staining, CD31+ endothelial cell networks, and quiescent stellate cells not expressing smooth muscle actin, which simulated the dynamic metabolism of drugs at the organ level. Although PHHs have strong metabolic capabilities, they rapidly lose their phenotype under 2D culture conditions [72]. Their long-term culture capacity is limited due to the shortage of healthy liver donors, resulting in a shortage of liver cells. After being induced with DMSO for two weeks, HepaRG cells have been shown to differentiate into liver- or bile-duct-like cells [73]. HepaRG cells demonstrate superior liver polarity, cell-to-cell connections, and metabolic functions in phases I and II, and they can form bile duct tubules [74]. Additionally, they express high levels of nuclear receptors and drug transporters. The HepaRG spheroid model is an appropriate 3D liver model for assessing and predicting liver toxicity.

#### 3.2.2. Hepatic Stellate Cells

Hepatic stellate cells play a crucial role in liver physiology, comprising 15% of all hepatocytes and 30% of all hepatic non-parenchymal cells [69]. Under normal conditions, these cells are in a dormant state. They are responsible for producing and releasing collagen and matrix components, such as glycoproteins and proteoglycans, and for regulating blood flow in the hepatic sinusoids. However, when triggered by specific physical, chemical, and biological factors, such as viral infection, hepatic stellate cells become activated, transform into “myofibroblasts”, and experience a significant rise in smooth muscle actin expression, collagen synthesis, and proliferation [75,76]. This phenomenon can be leveraged to create in vitro models of liver fibrosis. In addition, hepatic stellate cells have been shown to effectively imitate chronic liver injury in vivo [77].

#### 3.2.3. Hepatic Sinusoidal Endothelial Cells

Hepatic sinusoidal endothelial cells, located between the sinusoidal lumen and hepatocytes, play crucial roles in material transport, phagocytosis, and antigen presentation [78]. These cells have a distinctive structure, with a multi-window appearance, lacking a basement membrane and having high permeability that functions as a blood barrier. Additionally [79], they aid in hepatic extramedullary hematopoiesis by transporting stromal-derived factor (SDF)-1 to the bone marrow [80]. A study [36] showed that hepatic sinusoidal endothelial cells have the potential to support blood vessel formation by creating capillary-like buds in a printed hydrogel environment. These cells also regulate the extracellular matrix (ECM) turnover rate in the Disse space, serve as antigen recognition receptors [81], and interact with other immune cells to complete the immune response process.

#### 3.2.4. Kupffer Cells

Kupffer cells, also known as hepatic macrophages, serve as the body’s immune system’s primary defense against harmful particles and chemicals that enter the portal vein. These cells process and present antigens, enabling the body to mount an immune response [82]. Kupffer cells release a range of bioactive chemicals, including interleukin 6 (IL-6), interleukin-1 (IL-1), and hepatocyte growth factor (HGF), that are crucial for restoring the matrix of hepatocytes and regulating their function [83]. A study by Norona et al. [84] introduced Kupffer cells into a model to assess their impact on injury and fibrogenic responses following stimulation by cytokines and drugs. The results showed that the inclusion of Kupffer cells suppressed lactate dehydrogenase (LDH) levels, elevated the MTX-induced miR-122 factor, and delayed the expression of TGF-1. This study highlights the essential regulatory role that Kupffer cells play in liver fibrosis.

## 4. Bioprinted Liver Tissue Models

In recent years, there has been a surge in the creation of liver tissue models using bioprinting technology. This article summarizes the various models that are currently available. Table 2 provides a summary of existing 3D-bioprinted liver tissue models based on the method, cell sources, and primary research applications in hydrogel matrices.

### 4.1. Liver Tissue Models

#### 4.1.1. Scaffold-Based Models

Significant advances have been made in bioprinting livers [85], making it possible to produce functional liver tissues and mini-liver organs for drug research and liver tissue engineering. Yang et al. [86] used a 3D bioprinter to create in vitro liver tissue models with high structural integrity and high expression of liver-specific proteins and receptors, such as albumin, MPR2, CYP enzyme family, and glycogen, demonstrating the biological functions that the printed liver tissue should possess. These transplanted liver tissues have been shown to reduce liver injury and increase survival in mice with liver failure. Arai et al. [37] used the bionic hydrogel structure, which provides an enclosed 3D microenvironment with improved hepatocyte adhesion. It may be employed efficiently to create liver tissues for the long-term evaluation of cellular responses and drug analysis (Figure 2d).

The absence of a circulatory system has hindered the accurate replication of the liver in traditional tissue models. To address this challenge, creating a complex, perfusable vascular network is crucial for building realistic liver organ structures. Kang et al. [36] utilized low concentrations of GelMA and fibrin as bioink to print vascularized tissues with high cell viability and excellent resolution. The resulting macroscopic channels, endothelialized for bioprinting, allow for the creation of multiscale, vascularized liver tissues that can meet the centimeter-scale nutritional and oxygen demands of the liver (Figure 2e). Rania et al. [87] aimed to create an in vitro vascularized liver tissue that mimics the hepatic sinusoidal-like structure, using a coaxial extrusion-based bioprinting approach. This approach allowed for the study of cellular interactions between hepatocyte clusters and tubule-forming endothelial cells, leading to the development of a functional hepatic sinusoidal-like model.

#### 4.1.2. Scaffold-Free Models

Scaffold-free 3D culturing is another bioprinting technique that holds great promise for liver tissue research. One such method involves the formation of hepatocyte spheroids, aggregates with a diameter of 100–200 μm from individual hepatocytes [88]. These spheroids have been found to maintain their liver-specific functions and improve the hepatocyte survival time compared to traditional 2D cultures, making them a valuable in vitro model for liver metabolism and cytotoxicity research. The most commonly used method for creating these spheroid aggregates is the suspension drop technique [89], which involves dropping hydrogels containing functioning cells into a culture dish and allowing them to aggregate by gravity. This process creates spheroids of a predetermined size that can be maintained in culture for extended periods. Studies have shown that these spheroid aggregates resemble in vivo liver tissue at the proteome level and exhibit significant glycogen storage and albumin production, as well as robust expression of necessary metabolic and excretory enzymes, for up to two weeks [74]. Furthermore, researchers such as Elise et al. [90] have used droplets of hydrogel to construct primary human liver microtissues that can be cultivated for over nine days, and their vitality can be measured by relative adenosine triphosphate (ATP) concentration (Figure 3a). Liver microspheres created through scaffold-free 3D culturing techniques have proven to be effective for hepatotoxicity research. Despite being in culture for up to nine days, these microspheres have shown good cell viability, with high levels of proteomics and metabolomics. One other method of generating these spheroids is using cell rejection plates containing magnetic nanoparticles on the surface, which impede cell attachment and use magnetic force to form the spheroids quickly. This technique is suitable for hepatotoxicity research [91,92] (Figure 3b,c). Another approach, as demonstrated by Tostoes et al. [93], involves using an automated perfusion bioreactor to cultivate human hepatocyte spheroids that can be maintained for over four weeks, while retaining high levels of liver-specific markers, such as albumin, and CYP enzyme expression, as well as cell viability (Figure 3d). This makes them suitable for repeated-dose drug toxicity testing over long periods.

The construction and design of spheroids using either the hanging drop or cell exclusion plate method are closely tied to the desired functionality and purpose of cell cultures in vitro. Of particular note is the superior capability of scaffold-free hydrogel spheroids produced through 3D bioprinting technology when compared to traditional 2D cultures. These spheroids can regulate the spatial distribution of critical hepatocytes and decellularized extracellular matrix factors, replicate in vivo physiochemical microenvironments, facilitate the long-term culture of hepatocytes with physiologically relevant molecular phenotypes, and sustain liver-like functions similar to those found in vivo, all of which make them ideal for disease research and drug testing applications.

### 4.2. Disease Models

Modeling disease liver tissue is crucial in comprehending the underlying pathological mechanisms of liver diseases and evaluating the potential of drugs. The existence of hepatocellular carcinoma and liver fibrosis can gravely jeopardize human health; therefore, the establishment of a robust in vitro model is imperative for conducting in-depth research on the pathophysiology of liver diseases. These models can be used to assess the toxicity of different drug doses, simulate the disease state of the human liver in a laboratory setting for comprehensive drug screenings, improve the efficiency of pre-drug testing, and accurately predict clinical outcomes. This study examines the current models for liver cancer, liver fibrosis, and toxicity testing.

#### 4.2.1. Liver Cancer Models

Liver cancer is a primary global health concern that poses a significant threat to individuals. As per the 2021 statistics from the World Health Organization (WHO), it is the third most common cause of cancer-related deaths globally. Every year, approximately one million people are diagnosed with liver cancer worldwide [94]. Hepatocellular carcinoma is usually associated with liver cirrhosis, resulting from prolonged liver damage that may cause significant impairment of liver function [1]. Several factors, including alcohol-related hepatitis, viral hepatitis, and other liver disorders, can increase the likelihood of developing liver cancer [95]; hence, developing a reliable in vitro liver model is crucial for conducting extensive research on carcinogenesis and providing personalized treatment options. Mao et al. [96] used a composite hydrogel system made of gelatin-sodium alginate-matrix gel to print patient-derived primary tumor cells in a three-dimensional format. After a few days of culture, tumor markers, stem cell markers, fibrosis indices, and liver-specific protein expressions were measured. The tumor-related bioactivities and pharmacodynamic parameters were found to be more similar to those observed in patients compared to the results obtained from two-dimensional control cultures (Figure 4a). This 3D-printed model confirmed the primary tumor cells’ robust proliferation properties and invasive metastatic capacity and accurately replicated the original tumor’s critical biological and genetic features. Therefore, the in vitro liver cancer model provides a valuable tool for studying surrogate carcinogenesis and evaluating anti-cancer treatment resistance.

#### 4.2.2. Fatty Liver Model

The incidence of non-alcoholic fatty liver disease (NAFLD), the primary cause of metabolic liver disease leading to liver failure, has been on the rise in recent decades. To better understand the pathophysiology of NAFLD and its progression, it is crucial to develop reliable in vitro models for drug discovery and liver pathophysiology research [97]. To that end, a few microfluidic models have been created to study NAFLD, such as the one by Du et al. [98], who co-cultured HepG2 and HUVECS cells in gelatin spheres and transferred them to microfluidic chips. This model was then tested by exposing the cells to FFA-containing palmitic and oleic acids to stimulate the development of NAFLD and by measuring key metabolic parameters, such as albumin and urea secretion, CYP enzyme activity, and lipid droplet growth. Despite the promising results from these studies, there is still room for design and modeling improvement, given the small size of the liver chip [99] and the complex fabrication procedures required for each microstructure. Soon, it is expected that advances in 3D printing technology will lead to the creation of a 3D-printed fatty liver model.

#### 4.2.3. Liver Fibrosis Model

Various factors, such as viral hepatitis, fatty liver, and excessive alcohol consumption, can cause liver fibrosis. Despite its widespread prevalence and debilitating effects, there are no effective treatments for fibrosis, and current treatments mainly focus on managing symptoms [100]. Bioprinted liver fibrosis models have the potential to provide a platform for studying the disease progression and tissue restoration. Sofia et al. [101] co-cultured HepRG and HSC in 3D spheres in 96-well plates for 21 days. After 14 days of exposure to the pro-fibrotic chemicals propanolol and methotrexate, the liver models displayed fibrotic characteristics, including HSC activation, collagen secretion, and collagen deposition, demonstrating that the response to liver fibrosis involves multiple cells within the organ, not just single cell cultures (Figure 4b). In their study, Marie et al. [102] utilized a 3D-bioprinting-assisted photocuring method to embed HepaRG, LX-2, and HUVECs in GelMA and co-cultured them. They observed that liver cells in the co-culture system secreted TGFβ-1, which triggered the expression of ACTA2 and COL1A1 in LX-2 cells and subsequently led to the accumulation of fibrotic collagen. Furthermore, LX-2 cells exhibited a dormant state initially in the 3D culture model, and only after being treated with TGFβ-1 did they become activated. Until now, although researchers have managed to create the cell line successfully, the maximum survival duration of the current models is limited to just 14 days, which is inadequate for investigating the chronic toxicity of drugs. However, Marie et al. [102] have developed a model that can be maintained for up to four weeks, making it an excellent tool for drug testing and genetic toxicity studies (Figure 4c).

#### 4.2.4. Model of Drug-Induced Liver Damage

Before a new medication can be released to the market, it must undergo trial phases to assess its potential toxicity and predict the human response accurately. However, the existing model systems for toxicity prediction can be unreliable due to significant species differences, particularly in the specific expression of liver-related enzyme genes [77]. In contrast, bioprinted liver organoid models have demonstrated stable differentiation and proliferation for up to two weeks, with high cell viability and liver marker expression comparable to human liver cells [103] (Figure 4d). When exposed to an acetaminophen environment, the bioprinted liver-like organ model showed a decreased cell viability of 21–51% (*p* < 0.05) and elevated levels of the damage marker miR-122 in the culture medium, suggesting its potential as a substitute for human organs in toxicity testing [104]. Hence, the bioprinted liver-like organ model’s ability to predict toxicity and its standardization in creation makes it a promising candidate for toxicity testing.

**Figure 4 molecules-28-03683-f004:**
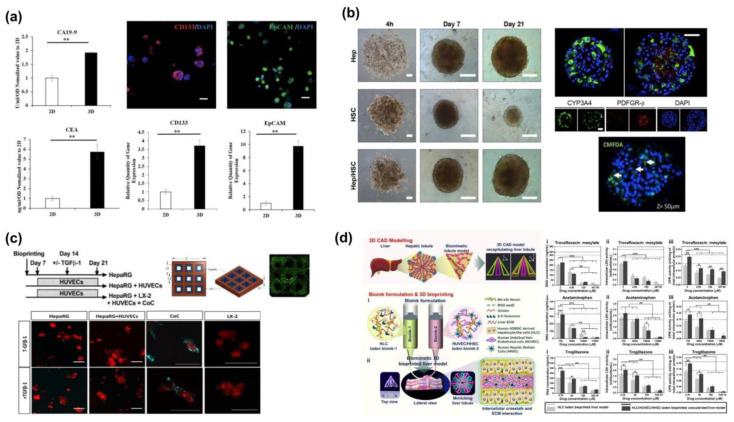
Existing disease models created using 3D bioprinting technology: (**a**) Expression of tumor biomarkers in ICC cells in 3D tumor models. Reprinted with permission from Ref. [96]. Copyright 2020 IOP; (**b**) Characteristics of co-cultivation of human 3D Hep/HSC spheroids. Reprinted with permission from Ref. [101]. Copyright 2016 Elsevier; (**c**) 3D-microengineered co-culture systems as models for fibrosis. Reprinted with permission from Ref. [102]. Copyright 2021 Elsevier; (**d**) Toxicity assessment of 3D bioprinted liver lobule microstructures against tevofloxacin mesylate, acetaminophen, and troglitazone to estimate DNA content, intracellular LDH activity, and CPR activity, * *p* ≤ 0.05, ** *p* ≤ 0.01, and *** *p* ≤ 0.001. Reprinted with permission from Ref. [103]. Copyright 2022 ACS.

**Table 2 molecules-28-03683-t002:** Summary of existing 3D-bioprinted liver tissue models.

Method	Cell Sources	Bioink	Application	Ref.
Extrusion-Based 3D Bioprinting	Fibroblast and hepatoma cells	Alginate and cellulose nanocrystals (CNCs)	A liver-mimetic honeycomb 3D structure	[22]
Extrusion-Based 3D Bioprinting	Hepatocytes; Human umbilical vein endothelial cells and human lung fibroblasts	Collagen with PCL	Angiogenesis for liver tissue engineering	[34]
Extrusion-Based 3D Bioprinting	Endothelial cells and mouse hepatocytes	dECM	A new dECM bioink	[59]
Extrusion-Based 3D Bioprinting	HepaRG	Gelatin and Alginate	Exploring the In Vivo and In Vitro Functionality of 3D-Printed Liver Organs	[86]
Extrusion-Based 3D Bioprinting	Hepatocytes and endothelial cells	GelMA	A Complex Liver Tissue with High Vascularization	[36]
Extrusion-Based 3D Bioprinting	ICC cells	Gelatin–Alginate–Matrigel	An intrahepatic cholangiocarcinoma tumor model	[96]
Inkjet-based 3D Bioprinting	Vascular endothelial cell	Gelatin and Alginate	Vascularization of tissue-engineered constructs	[26]
Inkjet-based 3D Bioprinting	Hepatocytes	Galactosylated alginate gel	New strategies for studying liver-specific functions of hepatocytes	[37]
Light-assisted 3D Bioprinting	HiPSC-HPCs; Human umbilical vein endothelial cells and Adipose-derived stem cells	GelMA and GMHA	A hexagonal liver lobule structure	[31]
Light-assisted 3D Bioprinting	Hepatic endothelial cells	GelMA and GMHA	A biomimetic hydrogel specifically designed to promote tissue repair	[49]
Light-assisted 3D Bioprinting	(hiPSC)-derived cardiomyocytes and hepatocytes	dECM	Striated heart and lobular liver structures	[35]
Light-assisted 3D Bioprinting	Hepatic parenchymal cells, HepaRG, with stellate cells (LX-2) and endothelial cells (HUVECs)	GelMA	A bioprinted liver model simulating hepatic fibrosis injury	[102]

## 5. Challenges and Shortcomings

The recent progress in developing 3D-printed liver tissue models showcases the growing interest in this transformative technology, quickly transitioning it from theoretical to practical applications and making it poised to be a leading technology shortly. In time, bioprinted livers have the potential to become a critical therapy option for liver disorders. However, more research and development are required to reach this goal. Although 3D printing technology has made it possible to create in vitro liver tissue models that closely mimic the in vivo environment, the microcosmic and macroscopic aspects of the human liver cannot be entirely bridged. As a result, there are still significant challenges in achieving fully functional liver tissue in vitro. The current limitations in bioprinting technology, such as the speed and resolution of bioprinting devices, need to be improved to create delicate and complex liver microstructures accurately and efficiently. The choice of bioink, the raw material used in 3D printing, is crucial for producing liver-like organ models. Currently, the development of personalized and customized liver models using bioprinting technology is in its early stages, and the bioinks must not only have good biocompatibility and stability, but also be safe for clinical applications and mimic the tissue-like structure of biological structures. As biotechnology and polymeric nanomaterials advance, new and improved biomaterials will be developed for medical research [105]. The bioprinted liver is still in the experimental phase and requires further research to ensure its safety and efficacy before it can be put into clinical use. Additionally, clinical trials will be necessary to validate its effectiveness in the treatment of liver diseases. It is important to consider the potential ethical implications of 3D-bioprinted livers, which may involve questions concerning ownership and consent surrounding the utilization of human tissue samples.

## 6. Conclusions

Bioprinting is a technology that uses a computer-controlled method to layer bioink-containing cells in order to precisely construct complex biological tissue and organ structures. This groundbreaking technique has been successfully used to develop various organ models, such as liver, skin, and heart models. At present, bioprinted liver models have been employed in researching liver cancer, liver fibrosis, and related areas [36,37,86,87]; however, the use of bioprinted liver tissue has not been widespread. As such, there is a pressing need to develop more diverse disease models, including non-alcoholic fatty liver, hepatitis, and several other disease models. Additionally, combining bioprinting with microfluidic organ chips can offer liver tissue models a complex, dynamic physicochemical microenvironment for growth and cultivation [106]. Overall, bioprinting of the liver is a highly significant field with tremendous potential to become a vital tool for researching and treating liver diseases, advancing the fields of precise drug therapy and complex organ tissue engineering to unprecedented heights. Further research is necessary to facilitate wider implementation and maximize its contributions to human health and medical research.

## Figures and Tables

**Figure 1 molecules-28-03683-f001:**
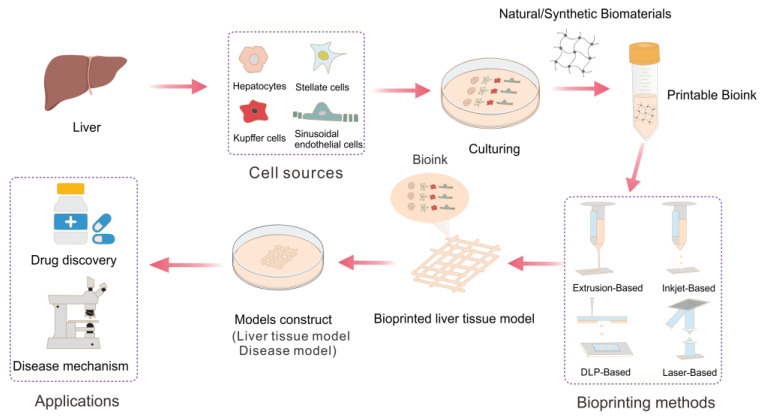
Procedure diagram for 3D bioprinting of liver tissue models and its application.

**Figure 2 molecules-28-03683-f002:**
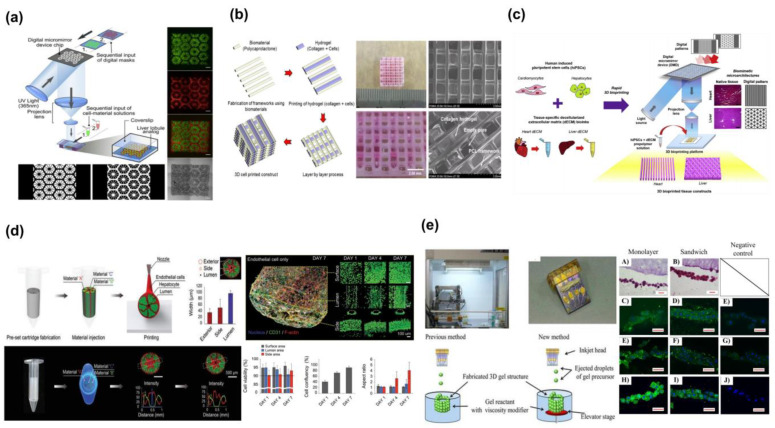
Existing liver scaffold-based models created using 3D printing technology: (**a**) Light-assisted-based hepatic construct. Reprinted with permission from Ref. [31]. Copyright 2016 PNAS; (**b**) A grid structure for cell deposition created using bioprinting with PCL scaffolds and collagen. Reprinted with permission from Ref. [34]. Copyright 2016 IOP; (**c**) Bioprinting of liver lobule structure using dECM. Reprinted with permission from Ref. [35]. Copyright 2019 Elsevier; (**d**) Predetermined extrusion bioprinting of a liver lobule structure. Reprinted with permission from Ref. [36]. Copyright 2020 Wiley Online Library; (**e**) A 3D-culture platform with sandwich architecture and its histological and immunofluorescent staining. Reprinted with permission from Ref. [37]. Copyright 2017 Wiley Online Library.

**Figure 3 molecules-28-03683-f003:**
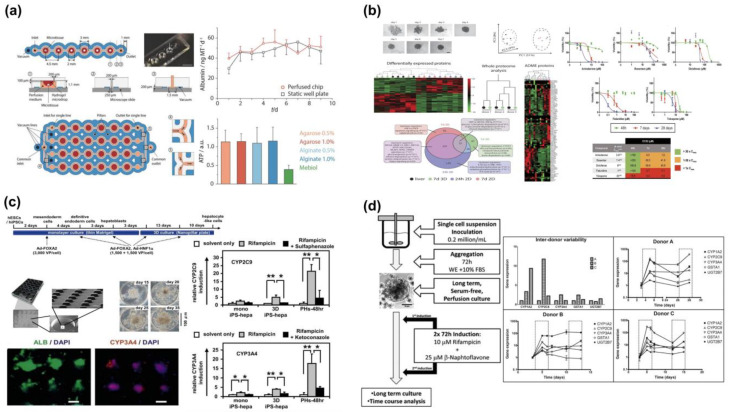
Existing liver scaffold-free models created using 3D printing technology: (**a**) Construction of primary human liver microtissues using droplet-based hydrogels. Reprinted with permission from Ref. [90]. Copyright 2018 Wiley Online Library; (**b**) 3D spheroids from PHH support chronic toxicity assays. Reprinted with permission from Ref. [91]. Copyright 2016 Nature; (**c**) 3D-cell spheroids derived from hESC/hiPSC-derived hepatocyte-like cells for hepatotoxicity detection, * *p* < 0.05; ** *p* < 0.01. Reprinted with permission from Ref. [92]. Copyright 2013 Elsevier; (**d**) Human liver cell spheroid for studying long-term liver metabolism and repeated-dose drug testing. Reprinted with permission from Ref. [93]. Copyright 2012 Wiley Online Library.

## Data Availability

Not applicable.

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
