# Peer review of "Advancements in Research on Constructing Physiological and Pathological Liver Models and Their Applications Utilizing Bioprinting Technology"

_molecules, 2023, doi:10.3390/molecules28093683_

Round 1

Reviewer 1 Report

1.       Review article

2.       Recommendation: Major revision

 Comments to Author:  

Ref. Molecules-2304264

The manuscript attempted to revise the advancements in 3D bioprinting technology in the field of liver tissue engineering. The topic looks good to the scientific community working in 3d printing and liver diseases.  Therefore, I recommend the publication of this manuscript with major revision because of the following reasons

1.      Keywords: should be arranged alphabetically

2.     Abstract part

·         The methodology part is not clear (how the semi-carbonized CDs were prepared is not clear)

·         The language should be improved.

·         The conclusion of the study is not well written based on the findings and should be rewritten

3.     Introduction

·         The first paragraph of the introduction, which states “The World Health Organization (WHO) reports that liver diseases affect 350 million individuals globally” is not correct, and is misleading. The absolute number of chronic liver disease cases (inclusive of any stage of disease severity) is estimated at 1.5 billion worldwide.

The authors can look at the following reference [Cheemerla, S., & Balakrishnan, M. (2021). Global Epidemiology of Chronic Liver Disease. Clinical liver disease, 17(5), 365–370. https://doi.org/10.1002/cld.1061].

The same is true on page 9 the last paragraph.

·         Page two the second paragraph the first statement is not clear! (Line numbers 50-51)

·         The actual advantage of 3D bioprinting of liver tissue is not well articulated.

·         The place for figure one is not the right place. I do not understand the importance of the figure in this topic!

Topics talking about the 3D Bioprinting technology from page 2-4 is simply a copy pest of from the following article  “Agarwal, T., Banerjee, D., Konwarh, R., Esworthy, T., Kumari, J., Onesto, V., ... & Ozbolat, I. T. (2021). Recent advances in bioprinting technologies for engineering hepatic tissue. Materials Science and Engineering: C, 123, 112013.]

Ø  The topic talking about hydrogels is not correct, the materials mentioned are all polymers (biomaterials) (natural, Synthetic, and semisynthetic) but not hydrogels. There is a big difference between hydrogel and biomaterials.  Indicating that the author has no knowledge about this topic.  The authors should rewrite.

Ø  All the figures are not clear and should be redrawn.

4.     Conclusion

The authors tried to address different biomaterials and different models for 3D printing of the liver; however, the conclusion lacks a comparison of the various biomaterials and different models to choose the best among them for researchers working in the area. The conclusion should be rewritten. 

Author Response

We express our gratitude for your expert review of our article. Your assessment has highlighted several areas that require attention. Based on your valuable recommendations, we have extensively revised our earlier manuscript, and the specific changes made are provided below. Below are the reviewer's comments, numbered for reference, and presented in regular font. Our responses are provided in red font, with any changes or additions made to the manuscript highlighted in red text.

  1. Keywords

Point 1: Keywords: should be arranged alphabetically.

Response 1: Thank you for your suggestion. As suggested by you, the keywords have been arranged alphabetically (Line 27).

  1. Abstract part

Point 2: The methodology part is not clear (how the semi-carbonized CDs were prepared is not clear).

Response 2: Thank you for your suggestion. I feel so sorry that my manuscript does not contain the part about how the semi-carbonized CDs were prepared.

Point 3: The language should be improved.

Response 3: Thank you for your valuable suggestion. We have tried our best to polish the language in the abstract part. (Lines 7-26).

Point 4: The conclusion of the study is not well written based on the findings and should be rewritten.

Response 4: Thank you for your valuable suggestion. The conclusion section has been rewritten based on the findings (Lines 23-26).

  1. Introduction

Point 5: The first paragraph of the introduction, which states “The World Health Organization (WHO) reports that liver diseases affect 350 million individuals globally” is not correct, and is misleading. The absolute number of chronic liver disease cases (inclusive of any stage of disease severity) is estimated at 1.5 billion worldwide.

The authors can look at the following reference [Cheemerla, S., & Balakrishnan, M. (2021). Global Epidemiology of Chronic Liver Disease. Clinical liver disease, 17(5), 365–370. https://doi.org/10.1002/cld.1061].

Response 5: We sincerely thank you for your careful reading. We may have made a mistake in our reading, and we apologize for our carelessness. We read the reference you provided carefully and found our problem. In our resubmitted manuscript, this data has been corrected and the appropriate reference has been inserted. Thank you for the correction (Line 30).

Point 6: The same is true on page 9 the last paragraph.

Response 6: We sincerely thank you for your careful reading. We replaced the correct data and added appropriate citation of reference citations to page 9, giving numbers 1 and 94-95 in the revised version of "References" (Page 11, Lines 443-448, respectively). 

Point 7: Page two the second paragraph the first statement is not clear! (Line numbers 50-51).

Response 7: We've rewritten the statement of the second paragraph on page two (Lines 55-57).

Point 8: The actual advantage of 3D bioprinting of liver tissue is not well articulated.

Response 8: Thank you for your suggestion. We have reiterated the practical advantages of utilizing 3D bioprinting technology for creating liver tissue (Lines 55-71).

Point 9: The place for figure one is not the right place. I do not understand the importance of the figure in this topic!

Response 9: Thank you for your suggestion. We have repositioned the Figure 1. Figure 1 primarily aims to demonstrate the sequential procedures necessary to produce a liver tissue model using 3D bioprinting (Lines 71-80).

Point 10: Topics talking about the 3D Bioprinting technology from page 2-4 is simply a copy pest of from the following article “Agarwal, T., Banerjee, D., Konwarh, R., Esworthy, T., Kumari, J., Onesto, V., ... & Ozbolat, I. T. (2021). Recent advances in bioprinting technologies for engineering hepatic tissue. Materials Science and Engineering: C, 123, 112013.]

Response 10: Thank you for your good suggestion. We regret the shortcomings in the writing of this section and have taken steps to improve it. Specifically, we have revised the discussion of 3D Bioprinting technology on pages 2-4 and have included relevant references to support our claims, which are assigned the numbers 20,23,24,27 and 32 respectively in the “References” in the revised version (Pages 2-4, Lines 82-87, 88-92 and 103-160).

Point 11: The topic talking about hydrogels is not correct, the materials mentioned are all polymers (biomaterials) (natural, Synthetic, and semisynthetic) but not hydrogels. There is a big difference between hydrogel and biomaterials. Indicating that the author does not know this topic. The authors should rewrite.

Response 11: We sincerely appreciate the valuable comments. We express our sorry for the lack of clarity regarding the distinction between hydrogels and biomaterials in our previous communication. To rectify this, we have thoroughly revised the topic and incorporated additional references for substantiation, which are assigned the number37,39,40-42,48 and 58-62respectively in the “References” in the revised version (Lines 177-206, 211-214, 220-227, 233-240 and 256-280).

Point 12: All the figures are not clear and should be redrawn.

Response 12: Thank you for your suggestion. We have redrawn all of the figures (Lines 71-80, 374-375, 420-421, and 509-510).

Point 13: The authors tried to address different biomaterials and different models for 3D printing of the liver; however, the conclusion lacks a comparison of the various biomaterials and different models to choose the best among them for researchers working in the area. The conclusion should be rewritten. 

Response 13: Thank you for your suggestion. According to your suggestion, we have rewritten the conclusion section (Lines562-576).

We have exerted our utmost effort to enhance the manuscript and have introduced modifications to the paper, which are marked in red, during the revision process. Rest assured, these changes do not have any impact on the paper's content or structure. We genuinely appreciate the reviewer's enthusiastic efforts and look forward to their approval of our revisions. Once again, we would like to express our sincere gratitude for their valuable opinions and suggestions.

Reviewer 2 Report

see attached pdf

Author Response

We express our gratitude for your expert review of our article. Your assessment has highlighted several areas that require attention. Based on your valuable recommendations, we have extensively revised our earlier manuscript, and the specific changes made are provided below. Below are the reviewer's comments, numbered for reference, and presented in regular font. Our responses are provided in red font, with any changes or additions made to the manuscript highlighted in red text.

Point 1: In lines 11-12, the word robust does not match with the phrase “more robust biomimetic advantages.” In terms of correct use of English. Please change the word. 

Response 1: We sincerely thank you for your careful reading. As suggested by you, we have rewritten the sentence and corrected the “robust” into “stronger” (Lines 13).

Point 2: In lines 50-60 and 64-76, add the following references and enrich the text by writing 2-3 sentences about 3D printing in regenerative medicine etc. 

  • 10.5923/j.mechanics.20211001.02 
  • 10.3844/ajeassp.2022.255.263 
  • 10.3390/pharmaceutics12020166 
  • 10.3390/ijms232314621 
  • 10.1016/j.promfg.2019.06.089 
  • 10.1016/j.compositesb.2018.02.012 
  • 10.1155/2019/5340616

Response 2: Thank you for your suggestion. We have written several sentences about 3D printing in regenerative medicine and added appropriate citations of references to enrich our article, which are assigned the numbers 8-10,12-13 and 15-16 respectively in the “References” in the revised version (Lines 57-61,85-87 and 92). 

Point 3: Try to restructure all tables so that they are not so wide, in order to look better in the Journal’s template.

Response 3: Thank you for your suggestion. We have restructured all of the tables in our article to improve their appearance and ensure they are properly formatted according to the journal's template (Lines 162-176 and 518-537).

Point 4: In lines 332-393, include some new figures because the plain text does not look very appealing for the readers. 

Response 4: Thank you for your valuable suggestion. We have prepared new figures and have added them to the manuscript as per your recommendation (Lines 420-421).

Point 5: Try to align figures, tables and text according to the Journal’s template, because that will create problems at the final editing.

Response 5: Thank you for your suggestion. We have aligned all of the figures, tables, and text in our manuscript to conform to the Journal's template (Lines 1-80, 162-176, 374-375, 420-421, 509-510, and 518-537).

Point 6: In the references list, try to include the DOI addresses of the articles in order to assist with their proper matching from the system.

Response 6: Thank you for your suggestion. We have considered your suggestion and included DOI addresses for the articles to facilitate their proper matching within the system (Lines 591-797).

We have exerted our utmost effort to enhance the manuscript and have introduced modifications to the paper, which are marked in red, during the revision process. Rest assured, these changes do not have any impact on the paper's content or structure. We genuinely appreciate the reviewer's enthusiastic efforts and look forward to your approval of our revisions. Once again, we would like to express our sincere gratitude for your valuable opinions and suggestions.

Round 2

Reviewer 1 Report

1.       Review article

2.       Recommendation: minor revision

 Comments to Author:  

Ref. Molecules-2304264 V2

1.       I am not still convinced with the place and legend for Figure 1. It can’t go with the topic. The explanation given in line number 70-71 is not properly written.

2.       The arrangement of topics still needs modification.

Generally, the authors tried to address most of the comments given, but the way they address looks not precise and to the point.1.       Review article

2.       Recommendation: minor revision

 Comments to Author:  

Ref. Molecules-2304264 V2

1.       I am not still convinced with the place and legend for Figure 1. It can’t go with the topic. The explanation given in line number 70-71 is not properly written.

2.       The arrangement of topics still needs modification.

Generally, the authors tried to address most of the comments given, but the way they address looks not precise and to the point.1.       Review article

2.       Recommendation: minor revision

 Comments to Author:  

Ref. Molecules-2304264 V2

1.       I am not still convinced with the place and legend for Figure 1. It can’t go with the topic. The explanation given in line number 70-71 is not properly written.

2.       The arrangement of topics still needs modification.

Generally, the authors tried to address most of the comments given, but the way they address looks not precise and to the point.1.       Review article

2.       Recommendation: minor revision

 Comments to Author:  

Ref. Molecules-2304264 V2

1.       I am not still convinced with the place and legend for Figure 1. It can’t go with the topic. The explanation given in line number 70-71 is not properly written.

2.       The arrangement of topics still needs modification.

Generally, the authors tried to address most of the comments given, but the way they address looks not precise and to the point.1.       Review article

2.       Recommendation: minor revision

 Comments to Author:  

Ref. Molecules-2304264 V2

1.       I am not still convinced with the place and legend for Figure 1. It can’t go with the topic. The explanation given in line number 70-71 is not properly written.

2.       The arrangement of topics still needs modification.

Generally, the authors tried to address most of the comments given, but the way they address looks not precise and to the point. 

Author Response

We would like to extend our appreciation for your expert review of our article. The reviewer's comments are provided below and presented in regular font. Our responses are indicated in red font, with any changes or additions made to the manuscript highlighted in red text.

Point 1:  I am not still convinced with the place and legend for Figure 1. It can’t go with the topic. The explanation given in line number 70-71 is not properly written.

Response 1: Thank you for your suggestion. I have revised Figure 1, repositioned it correctly, and added a detailed explanation. (Line 71-77).

Point 2: The arrangement of topics still needs modification. Generally, the authors tried to address most of the comments given, but the way they address looks not precise and to the point.

Response 2: Thank you for your suggestion. As suggested by you, the topics have been modified.

We sincerely appreciate the enthusiastic efforts of the reviewer and eagerly anticipate their approval of our revisions. We would like to express our deep gratitude for their valuable opinions and suggestions once again.

PS: The line numbers in the response letters correspond to the clear version of the manuscript.
